# Physical Self-Concept Changes in Adults and Older Adults: Influence of Emotional Intelligence, Intrinsic Motivation and Sports Habits

**DOI:** 10.3390/ijerph18041711

**Published:** 2021-02-10

**Authors:** Javier Conde-Pipó, Eduardo Melguizo-Ibáñez, Miguel Mariscal-Arcas, Félix Zurita-Ortega, Jose Luis Ubago-Jiménez, Irwin Ramírez-Granizo, Gabriel González-Valero

**Affiliations:** 1Department of Didactics of Musical, Plastic and Corporal Expression, University of Granada, Campus Universitario de Cartuja s/n, 18071 Granada, Spain; javiconde@correo.ugr.es (J.C.-P.); edumeliba@correo.ugr.es (E.M.-I.); felixzo@ugr.es (F.Z.-O.); irwinrg@ugr.es (I.R.-G.); ggvalero@ugr.es (G.G.-V.); 2Department of Nutrition and Food Science, University of Granada, Campus Universitario de Cartuja s/n, 18071 Granada, Spain; mariscal@ugr.es

**Keywords:** physical self-concept, psychosocial factors, older adults, physical activity, lifespan, aging

## Abstract

Lifespan is increasing globally as never before, and leading to an aging world population. Thus, the challenge for society and individuals is now how to live these years in the best possible health and wellbeing. Despite the benefits of physical activity for both are well documented, older people are not active enough. Physical self-concept is correlated with high levels of sports practice, although its evolution across one’s life span is not clear. The aim of this research has been to analyze the physical self-concept in older adults and its relationship with emotional intelligence, motivation and sports habits. The sample of 520 adults aged between 41 and 80 was clustered in ranges of age; 70.96% were men (*n* = 369; 57.34 years (SD: 7.97)) and 29.04% women (*n* = 151; age = 55.56 years (SD: 9.12)). Questionnaires adapted to Spanish were used to measure physical self-concept (Physical Self-Perception Profile), motivation (Sport Motivation Scale), and emotional intelligence (Trait Meta-Mood Scale). Regarding physical self-concept, the youngest group obtained the highest mean values and the oldest group the lowest. Physical self-concept correlated positively with emotional regulation and intrinsic motivation. Initiation to sports in childhood, the practice of sports activities for more than 150’ per week, and the practice of three or more sports, were associated with a higher score of physical self-concept. The findings reveal that physical self-concept declines in older adults, slightly at first, and sharply between 71 and 80 years, being intrinsic motivation, emotional regulation, and sports habits, factors to consider in favoring a positive physical self-concept and adherence to sporting activities.

## 1. Introduction

The world is experiencing a great demographic revolution. Facilitated by socioeconomic progress and the most recent advances in public health, life expectancy has increased significantly, which, together with a sharp drop in the birth rate, is leading to an aging world population. For the first time in history, the number of people aged over 65 is greater than under 5, and by 2050 the UN [1] expects this gap to double. Paradoxically, this great achievement of humanity presents new challenges, not only for society, which is seeing its workforce shrink or the pressure and costs of health and care services increase, but also for the individual who is faced with how to live through these times in the best possible health and quality of life [2,3,4,5].

It is now well established from a variety of studies that these problems can be mitigated by promoting a healthier and more active life among older people, an action that also impacts on quality of life, since the practice of physical activity (PA), in addition to reducing the risk of death from all causes, contracting cancer, diabetes or cardiovascular disease, has a direct impact on one of its main components, physical and mental functional capacity [4,6,7,8,9,10]. However, despite the many benefits of physical activity and the fact that the current older generation are healthier and feel younger than similarly aged individuals of past generations [6], older people are not active enough, with less than a third practicing PA regularly, so it is necessary to seek urgent measures to reverse this situation and increase participation [11,12,13,14].

The practice of PA also has benefits on the mental health of older people, positively affecting one of the most important psychosocial factors that influences an individual’s well-being and health habits, the self-concept [15,16,17,18,19]. The before-mentioned is defined as the collection of beliefs about one’s self and is also associated with health behaviors [15], and following the multidimensional and hierarchical model proposed by Shavelson [20], it is divided into academic, social, family, emotional and physical dimensions [15,21]. The physical self-concept (PSC) is therefore the set of ideas that we believe define us physically [22], and is in turn made up of four other dimensions, physical condition (PC), sports competence (SC), physical attractiveness (AT) and physical strength (ST) [23].

While the reciprocal relationship between PA and PSC is widely documented [12,24,25,26], the evolution of PSC across a lifespan is not clear. Longitudinal studies are still scarce, with comparisons between age-groups prevailing, mainly focusing on child and adolescent stages, and there is no consensus on the results [13,14,22,27,28,29]. In studies that have addressed this issue in adulthood and old age, the results are inconsistent. Some prominent longitudinal studies, such as Amesberguer et al. [30], Finkenzeller et al. [31], or Sweeney et al. [19], show that PSC is stable or slightly decreasing. Among the cross-sectional studies reviewed, Esnaola [32] reported an increase in PSC with age, while Infante et al. [33], Molero et al. [34], and Putnick et al. [35], reported that it remains stable.

Sports motivation (SM) plays an important role in the adherence to the practice of PA, determining the initiation, continuation and abandonment of it, since what pushes older individuals to practice is important, as is analyzing what factors determine it [36]. Currently, little is known about the specific reasons that might lead sedentary people to regular PA practice [37]. Despite the lack of studies on the older population, for which interest is growing, there is a body of recent research based on the theory of self-determination, which considers self-determined motivation as a predictor of positive psychological, social and emotional consequences in physically active older adults [38,39,40]. These authors suggest that it is mainly intrinsic motivation, such as the enjoyment or consideration of sports as a significant part of one’s lifestyle, that help the adherence to PA among the adult population, giving a secondary role to extrinsic motivation, such as competition or social recognition.

Emotional intelligence (EI) is another important factor that impacts on people’s social and psychological well-being [41,42]. As a personality trait, it encompasses emotion-related self-perceptions and dispositions, concerning how people manage emotions and understand the impact of their emotions on social interactions, and is measured via self-report [43,44]. Concerning the area of aging, research is still limited and inconsistent, although recent evidence suggests that EI is not stable over time, decreasing in some dimensions that can be compensated by increases in others, with a positive final balance [45,46,47,48]. Emotional regulation skills seem to increase with age due to personal experiences accumulated throughout life, and interpersonal relationships, generating greater emotional control, with less physiological activation in negative conditions and more positive emotional experiences [48,49]. However, due to age-specific cognitive decline, which leads to loss of attention, memory, and speed of information processing, older people may have more difficulty in accurately recognizing the emotions of others in facial gestures and vocabulary. They are also less expressive, making it more difficult to identify emotions in their faces [48,49,50].

There are numerous references found in the literature that describe the relationship between physical self-concept and sports habits, specifically the time of dedication to sports practice, as positive and reciprocal [12,18,24,25,26,51,52]. Nevertheless, the studies concerning the relationship between physical self-concept and intrinsic motivation are scarce, and their results are contradictory. Some of them consider this relationship to be positive [12,51,53], and others consider it negative [54]. Recent literature reviews published on the relationship between emotional intelligence and sports [55,56,57] show that a higher level of EI generates a more positive attitude towards the practice of PA, a greater frequency of practice, more fun and enjoyment, and a lower level of anxiety and competitive stress. Although the relationship with PSC has not yet been explored, indirectly, EI could contribute to higher levels of PSC.

Therefore, once the scientific literature has been reviewed, and the insufficiency of studies and the inconsistency of results have been proven, the aims of this research are the following: (a) to analyze the levels of physical self-concept in different age groups; (b) to establish the relationships between physical self-concept and sports motivation, emotional intelligence and sports habits, and (c) to examine the predictive capacity of the variables and physical self-concept in the Spanish population over the age of 40.

## 2. Materials and Methods

### 2.1. Design and Subjects

The study design was cross-sectional, descriptive, comparative and correlational. The sample was initially composed of 543 subjects, all of them Spaniards from different regions, with a single inclusion criterion, namely, being between 41 and 80 years old. In total, 23 participants were discarded for not meeting the age criteria or not completing the questionnaires properly. Of the 520 remaining (Table 1), 70.96% were men (*n* = 369; 57.34 years (SD: 7.97); height = 1.74 m (SD: 7.78); weight = 80.79 kg (SD: 11.69)) and 29.04% were women (*n* = 151; age = 55.56 years (SD: 9.12); height = 1.61 m (SD: 7.98); weight = 64.24 kg (SD: 13.13)). Following the statistical criteria, the sample was clustered in ranges of age. Thus, 22.30% of the subjects were between 41 and 50 years old (*n* = 116), 43.5% were between 51 and 60 years old (*n* = 225), 25.96% were between 61 and 70 years old (*n* = 135), and 8.24% were between 71 and 80 years old (*n* = 44).

### 2.2. Instruments and Variables

Ad-hoc questionnaire: this instrument was created by the researchers to collect the sociodemographic data and sports habits of the subjects participating in this research. They were asked about their sex, age, weight and height (from which the BMI was estimated), the hours of weekly sports practice, the age at which they began to practice sports, the number of sports practiced and which ones they were, grouped into outdoor sports (hiking, running, cycling, skiing, etc.), fitness sports (swimming, weights, pilates, yoga, dance, etc.) or opponents sports (tennis, paddle tennis, team sports). Based on the hours of weekly sports practice declared, all those who met the recently published recommendations for PA practice for older adults (≥150‘ per week) were labeled as “active” [6,58,59] and those who did not, as “not active”.

Physical Self-Concept Questionnaire (PSC-Q): this is based on the PSPP 30 (Physical Self-Perception Profile) questionnaire by Fox and Corbin [23] and was adapted to the adult Spanish population by Goñi [60]. It is made up of 30 items, and is valued using a Liker scale of four response options, from “Totally disagree” to “Totally agree”. It is divided into five dimensions: physical condition (PC, α = 0.84), sports competence (SC, α = 0.88), physical attractiveness (AT, α = 0.88), physical strength (ST, α = 0.83) and general physical self-concept (PSC, α = 0.88).

Sports Motivation Scale questionnaire (SMS): the original is by Pelletier [61], and it was adapted into Spanish by Balaguer et al. [62]. It is composed of three main dimensions, intrinsic motivation (IM, α = 0.79), extrinsic motivation (EM, α = 0.69) and demotivation (NM, α = 0.75), and consists of 28 items valued via a Likert scale that oscillates between values of 1 (“It has nothing to do with me”) and 7 (“It totally fits me”). It is an instrument with adequate psychometric properties, reliable and valid for the study of different types of motivational regulations in sport [36].

Emotional Intelligence Questionnaire (TMMS-24): the original is by Salovey et al. [63], and it was adapted to the Spanish language by Fernández-Berrocal et al. [64]. It assesses each person’s knowledge of their own moods using 24 items grouped in three dimensions: emotional attention (EA, α = 0.90), emotional understanding (EU, α = 0.90) and emotional regulation (ER, α = 0.86). It rates the responses using a Likert scale of 5 response options, from “Not at all agree” to “Strongly agree”.

### 2.3. Procedure

All subjects were recruited through social networks, participating voluntarily and showing their written informed consent. All of them were informed of the aims of this research and the data-protection provided was assured. The surveys were completed electronically and completely anonymously. The research complies with the principles of the Declaration of Helsinki and has the approval of the Research Ethics Committee of the University of Granada, with code 1230/CEIH/2020.

### 2.4. Analysis of Data

The statistical analysis was performed with the statistical computing software R (R Core Team, Vienna, Austria). For the basic descriptions, frequencies, means and standard deviations were used. For the study of comparisons between groups of continuous variables, the non-parametric tests for independent samples’ U of Mann–Whitney and Kruskal–Wallis were used. In the cases of lack of homoscedasticity and/or presence of outliers, the YUEN and robust ANOVA tests were chosen. The differences between groups were obtained with the Wilcoxon post hoc test with Bonferroni correction and the Lincon one. The level of effect in group comparisons was obtained using the standardized Cohen’s d and eta-squared indices, and in the case of bivariate correlations, Spearman’s rho. The association between categorical variables was evaluated using the chi-square test and its magnitude with Cramer’s V coefficient. The normality of data was assessed using the Kolmogoro–Smirnov test, using the Lillieforts correction, and homoscedasticity was assessed using the Levene test. The internal reliability of the instruments used was assessed using Cronbach’s alpha coefficient. All reported *p* values are based on a two-tailed test and the level of statistical significance set for all tests was 95%.

## 3. Results

Table 1 shows the characteristics of the sample according to classification by sex and age. Statistically significant differences were observed (*p* < 0.05) when comparing the variables of height and BMI by age groups, obtaining for both the greatest differences between the 41–50-year-old group and the 71–80-year-old group (*p* < 0.05). When comparing between sexes, statistically significant differences (*p* < 0.001) were obtained for the variables height, weight and BMI in all age groups, except for BMI in groups 61–70 (*p* = 0.175) and 71–80 (*p* = 0.537), and for weight in the 71–80 group (*p* = 0.100).

The sample’s sports habits by age ranges are shown in Table 2. A statistically significant relationship but weak association was found between age groups and sports initiation (*p* = 0.001; V = 0.171), level of activity (*p* = 0.001; V = 0.219), and number of sports practiced (*p* = 0.001; V = 0.188). The 41–50-year-old group was composed of a statistically higher percentage of active subjects (% = 93.97; *p* = 0.002), those initiated in childhood (% = 63.79; *p* = 0.004), and those practicing three or more sports (% = 55.17; *p* = 0.001). Contrastingly, the 71–80 group was the one with the highest percentage of non-active subjects (% = 29.55; *p* = 0.003) and non-practitioners (% = 15.91; *p* = 0.001), as well as the lowest percentage of practitioners of three or more sports (% = 25; *p* = 0.01). Regarding the type of sports practiced and the age groups, a significant but weak association was found both for maintenance sports (*p* = 0.046; V = 141) and for opponents (*p* = 0.001; V = 213), with statistically significant differences in the percentages of practitioners of each modality in the age group 41–50 (*p* ≤ 0.05).

The results of PSC-Q, TMMS-24 and SMS questionnaires are shown in Table 3. No dimensions presented a normal distribution (*p* > 0.05). The internal consistency analysis of PSC-Q showed acceptable values, both globally (α = 0.93) and for the different dimensions (SC, α = 0.83; PC, α = 0.87; AT, α = 0.81; ST, α = 0.72; PSC, α = 0.72). The means tests showed statistically significant differences between groups and weak association in the dimensions PSC (*p* = 0.022; η^2^ = 0.02), PC (*p* < 0.001; η^2^ = 0.03), and SC (*p* < 0.001; η^2^ = 0.03).

A different behavior was observed, whereby the youngest group obtained the highest mean values and the oldest group the lowest (Figure 1). For the PC dimension, the post hoc test with Bonferroni adjustment showed statistically significant differences between the 41–50 vs. 51–60 groups (*p* = 0.034) and the 41–50 vs. 71–80 groups (*p* = 0.039); for the SC dimension, the differences were between the 41–50 vs. 71–80 groups (*p* = 0.002) and the 61–70 vs. 71–80 groups (*p* = 0.043); for the PSC, all groups showed differences from the 71–80 group (vs. 41–50, *p* = 0.012; vs. 51–60, *p* = 0.034; vs. 61–70, *p* = 0.074).

No statistically significant differences (*p* > 0.05) were observed for the dimensions of the TMMS-24 questionnaire (α = 0.91), with the values remaining stable in the four age groups. The SMS questionnaire showed a global internal consistency of 0.93 (IM, α = 0.97; EM, α = 0.89; NM, α = 0.72), as well as statistically significant differences between groups and a mean association for the IM (*p* < 0.001; η^2^ = 0.06) and NM variables (*p* < 0.001; η^2^ = 0.06), placing the differences in the 71–80 group vs. the 41–50 (*p* = 0.003), 51–60 (*p* = 0.005) and 61–70 (*p* = 0.035) groups.

The analysis of the correlations between the questionnaires’ dimensions is presented in Table 4, all of them being positive except for NM. The correlation of all the scales of PSC-Q with each other was statistically significant and between moderate (r ≥ 0.5) and high (r ≥ 0.7), in contrast with those of SMS, which were small (r ≥ 0.1) to moderate, and those of TMMS-24 that ranged between null (r < 0.1) and moderate, finding no statistically significant correlation between EM and NM. The strongest correlations were obtained between PC and SC (r = 0.70), PC and ST (r = 0.64), and PC and AT (r = 0.60). The only dimension that did not correlate statistically significantly with the dimensions of other questionnaires was EA.

Table 5 shows the correlations between PSC and the rest of the PSC-Q, TMMS-24 and SMS variables by age group. The variables that present a statistically significant correlation with the PSC in all age groups are ST (r = 0.40; *p* ≤ 0.01), ER (r = 0.22; *p* ≤ 0.050) and NM (r = 0.32; *p* ≤ 0.002). The variables with the highest correlation with PSC were SC in groups 41–50 (r = 0.58; *p* < 0.001) and 51–60 (r = 0.62; *p* < 0.001), AT in 61–70 (r = 0.64; *p* < 0.001) and ST in 71–80 (r = 0.40; *p* = 0.010).

The association by age group between PSC and the sports habits studied is reflected in Table 6 and Figure 2. Concerning the stage of sports initiation, there were statistically significant differences in all age groups except for the 71–80 years old group (*p* = 0.180). Specifically, the greatest differences occurred between initiation during childhood and adulthood (41–50, *p* < 0.001; 51–60, *p* < 0.001; 61–70, *p* < 0.001). The strongest association was obtained in the 61–70 group (η^2^ = 0.23). No statistically significant differences were obtained for the variable level of activity in groups 41–50 (*p* = 0.362) and 71–80 (*p* = 0.262). In the groups in which it did occur, the association was large (51–60, *p* < 0.001, d = 0.50; 61–70, *p* < 0.001, d = 0.73). Regarding the number of sports practiced, an association with PSC was found in groups 51–60 (*p* < 0.001; η^2^ = 0.17) and 61–70 (*p* = 0.002; η^2^ = 0.24), with the greatest differences in those who practice three or more sports (*p* < 0.050) (Figure 3). No statistically significant differences were obtained depending on the type of sports practiced (*p* > 0.05).

## 4. Discussion

The main aim of this study has been to investigate the changes in PSC in the different age groups, covering the age range between 40 and 80 years, as well as its relationship with EI, SM and various sports habits.

Statistically significant differences in the behavior of PSC and its dimensions according to the age group were partially found. Despite the limitations of a cross-sectional study, the results obtained in relation to age in subjects over 40 indicate that the perception of PSC decreases with age, since a descending behavior is observed, especially in the older age group. However, statistically significant differences were only found for the group over 70 years of age, but not between the rest of the groups. Regarding the PC and SC dimensions, the differences occurred between the youngest and oldest age groups. As in other previous analogous studies, no statistically significant differences were found for the AT and ST dimensions [32,33,34], which indicates that the interest in both remains constant from the point of reaching adulthood.

These results are in line with those obtained in the few existing studies with the older population, both longitudinal [30,31] and transversal [32,33,34]. Based on this and previous research on PSC carried out with younger populations [32,60,65,66], we suggest that PSC throughout lifespan behaves in the shape of an inverted U, developing from infancy to the beginning of adulthood, followed by a period of stability until the age of 40–50, and then decreasing, first slowly, and sharply from the age of 70, concurring with the biological stage of the greatest loss of muscle strength, mobility and functional capacity [8,67,68]. The differentiated analysis of the PC and SC dimensions shows a behavior similar to the PSC but with a small rebound in the 61–70 age group, whose origin may be in the increase in free time to dedicate to sports and new physical activities after retirement, as well as the escape from the cessation of activity that this implies and its negative consequences for health [69]. Unlike the PSC, these dimensions could not stabilize, and fall from reaching their peak values in adolescence [32,60,70]. Therefore, reaching and maintaining high levels of PSC in advanced adults seems to depend on the levels reached during childhood and adolescence.

We also partially found a significant relationship between PSC and the dimensions that make up EI and SM in the different age groups, despite the close relationship between EI and its components with PA [56,71,72]. ER is the only dimension of EI that maintained a correlation with PSC in all age groups, probably due to its relationship with the ability to adjust levels of stress and anxiety [55,73], favoring positive emotional experiences and inhibiting the sensations of pain and fatigue [48,49,55]. However, the accumulated experience with age or the decline in cognitive abilities does not seem to influence the ability to identify, understand and manage emotions, as the average values of three EI dimensions remained stable when comparing the different age groups, contrary to what is stated by other studies [45,47,50,74]. The discrepancies in the results may be due, among other reasons, to the fact that our sample was limited to subjects aged over 40. However, the analysis of correlations between the different variables showed a negative correlation between NM and ER, which indicates that enhancing ER learning mechanisms can help prevent the abandonment of PA.

Regarding sports motivation, the IM dimension was positively correlated with the PSC except in the oldest group, and the NM dimension was negatively correlated in all groups. Regarding the dimensions of PSC, the highest correlation was obtained between IM and SC, which indicates that for this group of the population, control and dominance of the activity are important and motivating in themselves. In general terms, these results are congruent with those described in the existing literature and recently reviewed by Tang et al. [41], since they confirm the importance of intrinsic motivation and its correlation with psychological well-being and self-concept as a fundamental part of it. However, in the case of those over 71 years of age, PSC will depend to a greater extent on avoiding the loss of motivation and apathy towards sports practice than on the increase in personal satisfaction or mastery of the task, typical of intrinsic motivation.

Regarding the association between PSC and the sports habits contemplated in this study, the results obtained are in line with previous studies that have explored the relationship between the practice of PA and PSC [13,18,70,75]. In all age groups, the PSC level was higher in those who worked actively, obtaining the greatest differences in the group aged 61 to 70, possibly due to the slowing effect that physical activity has on the effects of aging, both physical and cognitive, that are very evident from this age onwards [16,76]. Regarding the type and number of sports practiced, although other studies with a younger population [77] found significant differences in the levels of PSC depending on the sport practiced, in our study these were not given, but they were regarding multidisciplinary practice. Those who practice three or more sports present higher levels of PSC in all age groups, with the differences between 50 and 69 years being statistically significant. Finally, an association was also found between the levels of PSC and the onset stage, this being higher in those sports subjects who started during the childhood stage, with the exception of the 71–80 group, possibly due to a generational effect, caused both by the lack of opportunities due to the socioeconomic context of the period and by the lack of school physical education. This finding adds one more reason to promote the practice of physical activity at an early age; not only will motor skills be developed and the foundations for optimal physical development be created, but higher levels of PSC will also be achieved that, if maintained in adulthood, will allow greater personal well-being and greater adherence to sports practice, which will ultimately translate into better physical and mental health [13,16,27].

This research has several limitations. The main one is its descriptive cross-sectional design, which does not allow us to establish a cause–effect relationship between the variables studied and the general PSC throughout older adults’ lifespans, requiring future longitudinal studies. Additionally, not all the population studied are older adults. The small sample size of both women and subjects over 70 may represent a bias when generalizing the results, and it would be interesting to expand it and complete the analysis differentiating by sex. Likewise, the absence of objective measurements for the assessment of physical activity, and the generational effect of each age group, make it necessary to take these results with prudence, and their interpretation must be cautious.

As a future perspective, the role of nutrition and other health parameters in the analysis of physical self-concept in the adult population should be considered.

## 5. Conclusions

The main finding of this research, despite its limitations, is that PSC decreases from the age of 40, with a sharp decline starting at the age of 70. Seen throughout the lifespan, and taking into account previous studies with younger populations, the behavior of PSC is not linear but convex, increasing from childhood until reaching adolescence, followed by a period of stability, and finally decreasing dramatically at the latter stages of life. Regarding the motivation towards sports practice and its relationship with PSC, it is in the intrinsic motivations where the older adult finds reasons for sports practice, thereby favoring higher levels of PSC. PA and higher levels of PSC are favored by EI through its ER dimension, as it reduces the effect of demotivation.

This research emphasizes the importance of practising physical activity from an early age, as well as multidisciplinary sports practice, since both are associated with higher levels of PSC in older adults. Not surprisingly, it is in childhood that motor skills and physical condition are acquired, and the more varied, stimulating and enriching these experiences are, the greater the enjoyment and adherence to PA practice will be, favoring it throughout the entire lifespan and bringing with it higher levels of health and quality of life.

## Figures and Tables

**Figure 1 ijerph-18-01711-f001:**
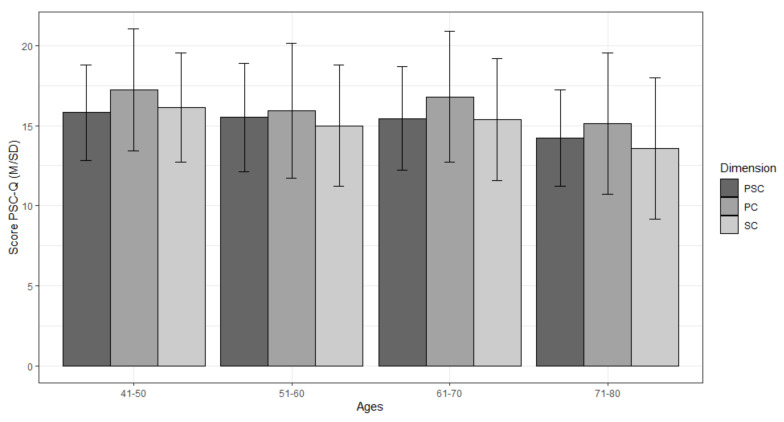
Trajectory of PSC, PC and SC with age.

**Figure 2 ijerph-18-01711-f002:**
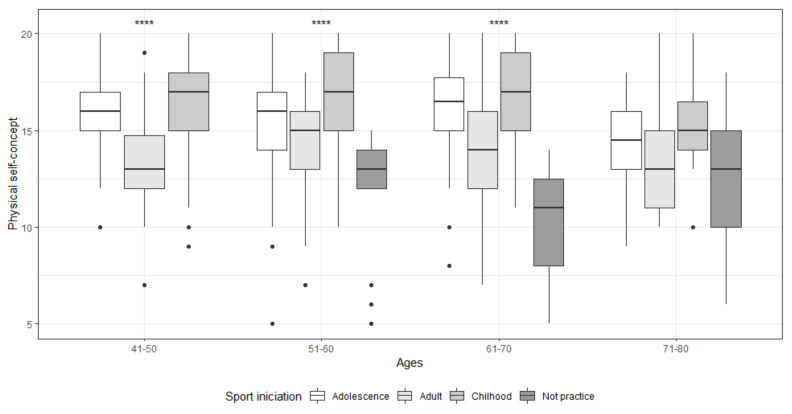
Association between PSC and sports initiation stage by age groups. **** Significatively at level 0.001. • Outliers.

**Figure 3 ijerph-18-01711-f003:**
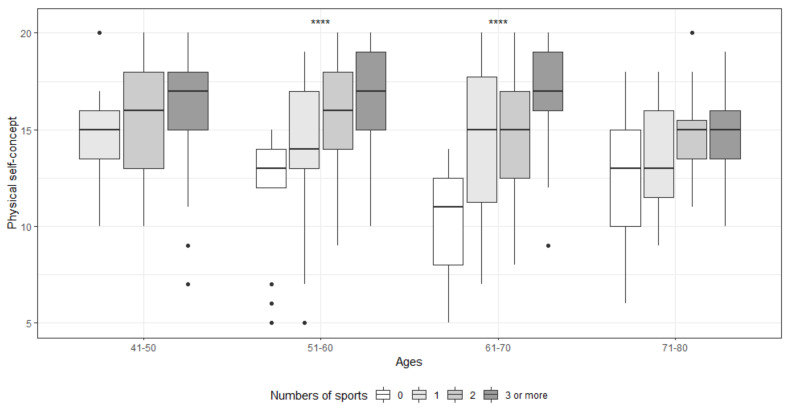
Relationship between PSC and number of sports practiced by age groups. **** Significatively at level 0.001. • Outliers.

**Table 1 ijerph-18-01711-t001:** Sample characteristics classified by age range and gender.

*n* = 520		41–50 (*n* = 116)			51–60 (*n* = 225)			61–70 (*n* = 135)			71–80 (*n* = 44)			
		M	W	p*_sex_* _*_	M	W	p*_sex_*_*_	M	W	p*_sex *_*	M	W	p*_sex *_*	p*_group **_*
Distribution	*n*	66	50	-	173	52	-	102	33	-	28	16	-	-
	%	56.89	43.10	0.167	76.88	23.11	0.001	75.55	24.44	0.001	63.63	36.36	0.006	0.005
Age (years)	M	45.58	45.50	0.952	55.69	55.29	0.461	63.21	63.39	0.688	73.86	71.81	0.007	0.001
	SD	2.55	2.74	-	2.68	2.87	-	2.73	2.83	-	2.90	2.83	-	-
	M_e_	46	46	-	56	56	-	62.5	63	-	73	70.5	-	-
	IQR	4.75	4.00	-	4.00	4.25	-	5	6	-	4.25	2.25	-	-
Height (m)	M	1.77	1.64	0.001	1.74	1.60	0.001	1.73	1.60	0.001	1.70	1.58	0.001	0.004
	SD	7.66	7.31	-	7.51	9.60	-	7.77	6.31	-	7.21	4.76	-	-
	M_e_	1.78	1.64	-	1.76	1.60	-	1.75	1.62	-	1.70	1.60	-	-
	IQR	10.5	9.5	-	9.00	9.25	-	10.0	8.5	-	7.5	5.5	-	-
Weight (kg)	M	82.69	62.8	0.001	81.46	62.52	0.001	79.69	67.1	0.001	75.78	69.09	0.100	0.149
	SD	12.55	15.13	-	12.13	9.19	-	10.82	14.64	-	8.00	12.31	-	-
	M_e_	79.0	59.5	-	80.0	60.0	-	78.0	63.0	-	77.0	68.0	-	-
	IQR	15.50	9.75	-	16.75	12.00	-	15.00	12.00	-	9.5	13.5	-	-
BMI (kg/m^2^)	M	26.31	23.30	0.001	26.72	24.52	0.001	26.69	26.00	0.175	26.23	27.71	0.537	0.002
	SD	4.19	5.46	-	4.36	4.23	-	4.33	5.55	-	3.22	5.34	-	-
	M_e_	25.76	21.82	-	26.07	22.96	-	25.71	25.31	-	25.14	27.09	-	-
	IQR	5.00	2.93	-	4.41	4.43	-	4.38	4.54	-	4.14	6.25	-	-

Note: M (man), W (woman). ** U Mann–Whitney*, ** Kruskal Wallis; IQM: “Inter Quartile Range”.

**Table 2 ijerph-18-01711-t002:** Sporting habits classified by age.

Variable	41–50		51–60		61–70		71–80		X^2^	
	*n*	%	*n*	%	*n*	%	*n*	%	*p*	V Cramer
Sports initiation									0.001	0.171
Childhood	74	63.79	113	50.22	57	42.22	16	36.36	-	-
Adolescence	20	17.24	64	28.44	42	31.11	12	27.27	-	-
Adult	22	18.97	35	15.56	30	22.22	9	20.45	-	-
Not practice	0	0.00	13	5.78	6	4.44	7	15.91	-	-
PA level									0.001	0.219
Not active	7	6.03	45	20.00	22	16.30	13	29.55	-	-
Active	109	93.97	180	80.00	113	83.70	31	70.45	-	-
Number of sports practiced									0.001	0.188
None	0	0.00	13	5.78	6	4.44	7	15.91	-	-
1	11	9.48	35	15.56	26	19.26	11	25.00	-	-
2	41	35.34	100	44.44	47	34.81	15	34.09	-	-
3 or more	64	55.17	77	34.22	56	41.48	11	25.00	-	-
Sports										
Outdoor	87	75.00	160	71.11	94	69.63	28	63.64	0.366	0.089
Fitness	68	58.62	95	42.22	67	49.63	18	40.91	0.046	0.141
Opponent	31	26.72	39	17.33	22	16.30	2	4.55	0.001	0.213

**Table 3 ijerph-18-01711-t003:** Results of the PSC, TMMS-24 and SMS questionnaires and association with age.

Variable		41–50	51–60	61–70	71–80	Levene’s Test		Means’s Test		Effect Size	
		M(SD)	M(SD)	M(SD)	M(SD)	F	*p*	X^2^	*p*	η^2^	IC 95%
PSC-Q	PSC	15.81(2.97)	15.51(3.40)	15.45(3.24)	14.22(3.01)	0.775	0.508	9.571	0.022	0.02	(0.00, 0.04)
	PC	17.25(3.80)	15.94(4.21)	16.80(4.07)	15.13(4.42)	1.117	0.341	13.326	0.001	0.03	(0.00, 005)
	AT	21.93(5.05)	22.31(4.72)	22.82(4.79)	22.04(3.71)	2.225	0.084	2.955	0.398	0.00	(0.00, 0.01)
	SC	16.13(3.43)	15.00(3.80)	15.40(3.81)	13.59(4.40)	2.239	0.084	16.344	0.001	0.03	(0.01, 005)
	ST	14.25(2.76)	14.12(3.08)	14.10(3.38)	13.79(3.23)	1.265	0.285	1.300	0.279	0.00	(0.00, 0.00)
TMMS24	EA	23.99(6.86)	23.96(6.88)	23.00(6.66)	23.20(7.66)	0.476	0.698	2.343	0.504	0.00	(0.00, 0.00)
	EU	26.19(6.88)	27.57(6.46)	26.52(5.73)	25.54(6.09)	1.878	0.132	4.485	0.213	0.01	(0.00, 0.03)
	ER	28.12(6.10)	28.67(6.48)	27.34(6.44)	27.45(6.58)	0.729	0.535	4.245	0.236	0.01	(0.00, 002)
SMS	IM	5.71(1.09)	5.48(1.42)	5.24(1.47)	4.39(1.79)	4.803	0.002	5.999	0.001	0.06	(0.02, 0.09)
	EM	3.82(1.08)	3.89(1.19)	3.76(1.35)	3.72(1.42)	8.647	0.034	0.687	0.529	0.04	(0.01, 0.07)
	NM	2.15(1.13)	2.08(1.08)	2.45(1.24)	2.88(1.47)	3.692	0.011	5.717	0.001	0.04	(0.01, 0.07)

Note: PSC (physical self-concept), PC (physical condition), AT (physical attractiveness), SC (sports competence), ST (strength), EA (emotional attention), EU (emotional understanding), ER (emotional regulation), IM (intrinsic motivation), EM (extrinsic motivation), NM (demotivation).

**Table 4 ijerph-18-01711-t004:** Bivariate correlations between the dimensions of the PSC-Q, TMMS and EMD questionnaires.

Dimension	PSC		ST		SC		AT		PC		EA		EU		ER		IM		EM	
ST	0.54	**	1																	
	(0.47, 0.60)																			
SC	0.58	**	0.60	**	1															
	(0.52, 0.63)		(0.54, 0.65)																	
AT	0.56	**	0.57	**	0.49		1													
	(0.50, 0.63)		(0.51, 0.63)		(0.43, 0.56)															
PC	0.51	**	0.64	**	0.70	**	0.60	**	1											
	(0.45, 0.57)		(0.58, 0.69)		(0.65, 0.74)		(0.55, 0.65)													
EA	0.02		0.06		0.02		0.01		0.06		1									
	(−0.06, 0.11)		(−0.03, 0.15)		(−0.07, 0.11)		(−0.08, 0.09)		(−0.03, 0.15)											
EU	0.15	**	0.19	**	0.16	**	0.11	*	0.16	**	0.33	**	1							
	(0.07, 0.24)		(0.10, 0.27)		(0.08, 0.24)		(0.02, 0.19)		(0.08, 0.24)		(0.25, 0.40)									
ER	0.31	**	0.30	**	0.29	**	0.25	**	0.31	**	0.21	**	0.48	**	1					
	(0.23, 0.38)		(0.22, 0.38)		(0.21, 0.36)		(0.17, 0.33)		(0.23, 0.38)		(0.13, 0.29)		(0.41, 0.54)							
IM	0.38	**	0.35	**	0.54	**	0.26	**	0.45	**	0.16	**	0.29	**	0.35	**	1			
	(0.30, 0.45)		(0.28, 0.43)		(0.48, 0.60)		(0.18, 0.34)		(0.38, 0.52)		(0.08, 0.24)		(0.21, 0.36)		(0.27, 0.42)					
EM	0.14	**	0.16		0.38	**	0.13	**	0.32	**	0.14	**	0.08		0.17	**	0.48	**	1	
	(0.05, 0.22)		(0.07, 0.24)		(0.31, 0.46)		(0.05, 0.21)		(0.24, 0.39)		(0.06, 0.22)		(−0.01, 0.16)		(0.08, 0.25)		(0.41, 0.54)			
NM	−0.38	**	−0.26	**	−0.28	**	−0.27	**	−0.28	**	0.00		−0.25	**	−0.25	**	−0.36	**	0.07	**
	(−0.45, −0.31)		(−0.34, −0.26)		(0.36, −0.20)		(−0.35, −0.19)		(−0.36, −0,20)		(0.09, 0.08)		(−0.33, −0.16)		(−0.33, −0.17)		(−0.44, 0.29)		(0.01, 0.07)	

Note 1: PSC (physical self-concept), PC (physical condition), AT (physical attractiveness), SC (sports competence), ST (strength), EA (emotional attention), EU (emotional understanding), ER (emotional regulation), IM (intrinsic motivation), EM (extrinsic motivation), NM (demotivation). Note 2: ** (*p* ≤ 0.001), * (*p* ≤ 0.010).

**Table 5 ijerph-18-01711-t005:** Correlations between PSC and the dimensions of the PSC-Q, TMMS and EMD questionnaires by age groups.

Dimension	41–50 Years			51–60 Years			61–70 Years			71–80 Years		
	r	IC	*p*	r	IC	*p*	r	IC	*p*	r	IC	*p*
ST	0.51	(0.36, 0.51)	0.001	0.54	(0.44, 0.63)	0.001	0.56	(0.43, 0.67)	0.001	0.40	(0.12, 0.63)	0.010
SC	0.58	(0.44, 0,69)	0.001	0.62	(0.53, 0.69)	0.001	0.57	(0.45, 0.68)	0.001	0.20	(−0.10, 0.47)	0.190
AT	0.50	(0.35, 0.63)	0.001	0.58	0.49, 0.66)	0.001	0.64	(0.53, 0.73)	0.001	0.28	(−0.02, 0.53)	0.060
PC	0.51	(0.36, 0.51)	0.001	0.52	(0.42, 0.61)	0.001	0.60	(0.48, 0.70)	0.001	0.23	(−0.07, 0.49)	0.140
EA	−0.21	(−0.38, −0.21)	0.020	0.05	(−0.08, 0.18)	0.430	0.09	(−0.08, 0.26)	0.280	0.13	(−0.18, 0.41)	0.420
EU	0.17	(−0.02, 0.17)	0.070	0.16	(0.03, 0.28)	0.001	0.07	(−0.10, 0.24)	0.420	0.22	(−0.08, 0.49)	0.140
ER	0.26	(0.08, 0.42)	0.010	0.37	(0.25, 0.48)	0.001	0.22	(0.06, 0.38)	0.010	0.30	(0.01, 0.55)	0.050
IM	0.26	(0.08, 0.42)	0.001	0.46	(0.35, 0.56)	0.001	0.34	(0.18, 0.48)	0.001	0.25	(−0.05, 0.51)	0.100
EM	0.12	(−0.02, 0.34)	0.190	0.21	(0.08, 0.33)	0.001	0.06	(−0.11, 0.23)	0.500	0.07	(−0.23, 0.36)	0.650
NM	−0.42	(0.56, −0.25)	0.001	−0.39	(−0.50, −0.28)	0.001	−0.32	(−0.46, −0.32)	0.001	−0.34	(−0.58, −0.05)	0.020

**Table 6 ijerph-18-01711-t006:** Association between PSC and sports habits by age groups.

Variable	41–50 Years			51–60 Years			61–70 Years			71–80 Years		
	*p*	η^2^	IC	*p*	η^2^	IC	*p*	η^2^	IC	*p*	η^2^	IC
Sportsinitiation	0.001	0.19	(0.08, 0.29)	0.001	0.19	(0.11, 0.26)	0.001	0.23	(0.13, 0.32)	0.180	0.12	(0.00, 0.26)
	M		SD	M		SD	M		SD	M		SD
Not practice	-		-	11.62		3.33	10.17		3.49	12.43		4.20
Childhood	16.53		2.46	16.53		2.46	16.53		2.44	15.38		2.71
Adolescence	15.80		2.59	15.23		3.13	15.98		2.82	14.25		2.45
Adult	13.41		2.68	14.20		2.46	13.77		3.13	13.56		3.09
PA level	*p*	d	IC	*p*	d	IC	*p*	d	IC	*p*	d	IC
	0.362	0.20	(0.56, 0.96)	0.001	0.50	(0.17, 0.83)	0.005	0.73	(0.26, 1.19)	0.262	0.51	(0.15, 1.17)
	M		SD	M		SD	M		SD	M		SD
Not active	15.28		1.97	14.33		2.91	13.54		3.77	13.15		3.33
Active	15.84		2.81	15.81		2.94	15.83		3.01	14.67		2.80
Number of sports	*p*	η^2^	IC	*p*	η^2^	IC	*p*	η^2^	IC	*p*	η^2^	IC
	0.173	0.02	(0.00, 0.07)	0.001	0.17	(0.09, 0.24)	0.001	0.24	(0.13, 0.33)	0.401	0.10	(0.00, 0.22)
	M		SD	M		SD	M		SD	M		SD
None	-		-	11.62		3.33	10.17		3.49	12.43		4.20
1	14.82		2.79	14.37		3.21	14.58		3.78	13.73		3.10
2	15.59		2.79	15.58		2.63	14.81		2.74	15.07		2.55
3 or more	16.12		2.75	16.61		2.55	16.98		2.39	14.73		2.45
Sports	*p*	η^2^	IC	*p*	η^2^	IC	*p*	η^2^	IC	*p*	η^2^	IC
	0.531	0.01	(0.00, 0.03)	0.543	0.19	(0.00, 0.01)	0.914	0.00	(0.00, 0.01)	0.666	0.02	(0.00, 0.10)
	M		SD	M		SD	M		SD	M		SD
Outdoors	15.28		2.93	15.91		2.66	16.02		2.86	14.96		2.47
Fitness	15.72		2.90	16.17		2.88	16.19		2.83	14.39		2.89
Opponent	16.39		2.62	15.87		2.93	16.00		3.01	13.50		2.12

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
