# Peer review of "Physical Self-Concept Changes in Adults and Older Adults: Influence of Emotional Intelligence, Intrinsic Motivation and Sports Habits"

_ijerph, 2021, doi:10.3390/ijerph18041711_

Round 1

Reviewer 1 Report

The topic is nice as aging population is increasing in the world. Physical activity is beneficial to everyone, especially the older adults to lower the mortality. But I have some comments about the manuscript as follows.

  1. The authors used the 4-branch emotional intelligence (EI) model. However, the TMMS-24 measured 3 dimensions, which was not consistent with the background of EI.
  2. What is the reliability and validity of Trait Meta-Mood Scale (TMMS-24)?
  3. Regarding the name of TMMS, which definition of EI was using in this study, as a trait or ability? Trait EI is rather stable over ages, while ability EI could change over times.
  4. The World Health Organization suggests adults should do exercise more than 150' per week. Please explain the reason of using 120' as 'active' in this study.
  5. It would be better to show the hypothesis in a diagram.

Author Response

Dear reviewer,

We appreciate the time taken to review this manuscript and the comments made, which we believe to be critical for producing rigorous and quality research. Below are detailed all the changes made in the original article: ijerph-1104149

Amendments have been made in the original manuscript following reviewer's comments. Changes made are colored to be easily identified.

MODIFICATIONS

Comment 1.

The topic is nice as aging population is increasing in the world. Physical activity is beneficial to everyone, especially the older adults to lower the mortality.

Response 1.

Authors are grateful for your comments, for evaluating the work, and considering this topic of interest.  

Comment 2.

The authors used the 4-branch emotional intelligence (EI) model. However, the TMMS-24 measured 3 dimensions, which was not consistent with the background of EI.

Response 2.

Authors are grateful for your comment, which has allowed us to clarify the introduction and, as you require, give it consistency.

Indeed, the TMMS scale is based on the previous 3-branch EI model, enunciated by Salovey and Mayers (1990), and not on the current model of these same authors (4 Branch). However, both models are still valid and there are many investigations on EI carried out with the TMMS instrument, especially in the Spanish language, reasons for which it was selected by the authors, in addition to evaluate, steadily in time, people’s beliefs about their mood states and emotions, making it ideal for this research.

In any case, as suggested by Fernández-Berrocal and Extremera (2008), it is necessary for future research to take into account the need to adapt the TMMS factors to the dimensions of the new EI model proposed by Salovey and Mayers (1995 ), and we have reflected this in the future perspectives of this research.

Comment 3.

What is the reliability and validity of Trait Meta-Mood Scale (TMMS-24)?

Response 3.

Authors are grateful for your question.  TMMS-24 reliability has been added in section Instruments and variables: emotional attention  (EA,  α = 0.90), emotional understanding (EU, α = 0.90) and emotional regulation (ER,  α = 0.86).

Comment 4.

Regarding the name of TMMS, which definition of EI was using in this study, as a trait or ability? Trait EI is rather stable over ages, while ability EI could change over times.

Response 4.

Authors are grateful for your question. The definition used by the authors in this study is as trait, precisely because it is, as you appreciate, more stable over time.

Comment 5.

The World Health Organization suggests adults should do exercise more than 150' per week. Please explain the reason of using 120' as 'active' in this study.

Response 5.

Authors are grateful for your comment. Indeed, the recommendation for physical activity from the WHO and another important institution in public health, such as the American College of Sport Medicine, is 150 'per week, and this has been taken into account for the development of this research. We made an unfortunate typing error that has been changed in both the abstract and the instruments section. We have also updated the references including the latest WHO recommendations (2020).

Comment 6

It would be better to show the hypothesis in a diagram.

Response 6.

Authors are grateful for your comment. Following the request of reviewer 2, the hypothesis have been eliminated.

Reviewer 2 Report

The objective of the study was to analyze physical self-concept in older adults and its relationship with emotional intelligence, motivation and sports habits. The sample included 520 adults from 41 to 80 years. 

I consider this is an interesting topic. However, in order to be accepted for publication, major changes must be addressed:

1) I suggest improving scientific English and changes in structure throughout the manuscript. For example:

Remove the hypotheses from the text. 

In methods and results sections "Quartiles" of age is used for the age ranges (set by the authors). The term Quartiles usually is used for values that divide the data into quarters.

Sample characteristics can be described as part of the results and not in the methods section. I recommend changing Table 1 to the results section. Also, this table shows means and SDs for variables. If Kruskal Wallis Test was used, I suggest the inclusion of medians and Interquartile ranges for skewed variables. 

Revise the word "sport" in all the sections of the manuscript, I highly suggest to revise and change for "sports", i.e. when you refer as sports habits or sports practice.

Did you obtain data on chronic diseases or disability in subjects? These variables could be of great importance on physical activity and self-perception scales. If you have data on such parameters, correlations need to be adjusted for such covariates.

As stated: "The differences between groups were obtained with the Wilcoxon post hoc test with Bonferroni correction and the Lincon one" ¿Could you provide a reference to "Lincon" correction? 

Please provide conclusions according to the results and avoid speculation.

2) Correct all references. In the current manuscript, numbers appear in random order. References must be numbered in order of appearance in the text (including table captions and figure legends) and listed individually at the end of the manuscript. 

Author Response

Dear reviewer,

We appreciate the time taken to review this manuscript and the comments made, which we believe to be critical for producing rigorous and quality research. Below are detailed all the changes made in the original article: ijerph-1104149

Amendments have been made in the original manuscript following reviewer's comments. Changes made are colored to be easily identified.

MODIFICATIONS

Comment 1.

The objective of the study was to analyze physical self-concept in older adults and its relationship with emotional intelligence, motivation and sports habits. The sample included 520 adults from 41 to 80 years.

I consider this is an interesting topic. However, in order to be accepted for publication, major changes must be addressed:

Response 1.

Authors are grateful for your comments, for evaluating the work, and considering this topic of interest.  

Comment 2.

1) I suggest improving scientific English and changes in structure throughout the manuscript. For example:

Remove the hypotheses from the text.

Response 2.

Authors are grateful your comment. The manuscript has been revised again and improved it,  by a English native who is Sport Sciences degree. Following your request, hypotheses has been removed.

Comment 3.

In methods and results sections "Quartiles" of age is used for the age ranges (set by the authors). The term Quartiles usually is used for values that divide the data into quarters.

Response 3.

Authors are grateful for your comment. The term Quartiles has been removed, finally remaining as follows “ the sample was clustered in range of age”.

Comment 4.

Sample characteristics can be described as part of the results and not in the methods section. I recommend changing Table 1 to the results section. Also, this table shows means and SDs for variables. If Kruskal Wallis Test was used, I suggest the inclusion of medians and Interquartile ranges for skewed variables.

Response 4.

Authors are grateful for your comment . Following your suggestions, tabla1 has been moved to results section and  medians and interquartile ranges adds.

Comment 5.

Revise the word "sport" in all the sections of the manuscript, I highly suggest to revise and change for "sports", i.e. when you refer as sports habits or sports practice.

Response 5.

Authors are grateful for your comment. The manuscript has been revised and modified sport for sports.

Comment 6.

Did you obtain data on chronic diseases or disability in subjects? These variables could be of great importance on physical activity and self-perception scales. If you have data on such parameters, correlations need to be adjusted for such covariates.

Response 6.

Authors are grateful for your question. No health variable was obtained, so the authors have included this point in a future perspective.

Comment 7.

As stated: "The differences between groups were obtained with the Wilcoxon post hoc test with Bonferroni correction and the Lincon one" ¿Could you provide a reference to "Lincon" correction?

Response 7.

Authors are grateful for your question. Let us clarify than Lincon is a post hoc test for robust ANOVA,  not a correction of Wilcoxon test. Authors have changed  the phrase to avoid confusions.

Reference:

Mair P, Wilcox RR (2020). “Robust Statistical Methods in R Using the WRS2 Package.” Behavior Research Methods, 52, 464–488.

Comment 8.

Please provide conclusions according to the results and avoid speculation.

Response 8.

Authors are grateful for your comment. Certainly, some conclusions were based in speculation and have been removed.

Comment 9.

2) Correct all references. In the current manuscript, numbers appear in random order. References must be numbered in order of appearance in the text (including table captions and figure legends) and listed individually at the end of the manuscript.

Response 9.

Authors are grateful for your comment. The manuscript was send in free format. Author have changed format to adapt it to IJERPH.

Reviewer 3 Report

The submitted manuscript has numerous valuable contributions to older people's research. It also raises some questions and suggestions for the improvement of the article.

I suggest that the title be changed to the following: "Physical self-concept changes in adults and older adults: influence of emotional intelligence, intrinsic motivation and sports habits."

In the introduction section a small paragraph should be added specifying the influence of emotional intelligence, intrinsic motivation and sports habits on physical self-concept.

In the material and methods section it is necessary that the authors provide the code of ethics committee.

The results section is sufficiently scientifically sound and precisely detailed.

In the discussion section, the authors should point out as a limitation that not all the population studied are older adults.

As for the references, there are some errors in the adaptation to the regulations. In addition, it could include some more current references on this topic. Please revise.

Author Response

Dear reviewer,

We appreciate the time taken to review this manuscript and the comments made, which we believe to be critical for producing rigorous and quality research. Below are detailed all the changes made in the original article: ijerph-1104149

Amendments have been made in the original manuscript following reviewer's comments. Changes made are colored to be easily identified.

MODIFICATIONS:

Comment 1

The submitted manuscript has numerous valuable contributions to older people's research. It also raises some questions and suggestions for the improvement of the article.

Response 1

Authors are grateful for your comment and for appraising  this manuscript, which, thanks to all reviewers indications, has been notably improved

Comment 2

I suggest that the title be changed to the following: "Physical self-concept changes in adults and older adults: influence of emotional intelligence, intrinsic motivation and sports habits."

response 2

Authors are grateful for your suggestion. The title has been changed, we agree that the one proposed by you is more appropriate.

Comment 3

In the introduction section a small paragraph should be added specifying the influence of emotional intelligence, intrinsic motivation and sports habits on physical self-concept.

Response 3.

Authors are grateful for your comment. A paragraph about the influence of intrinsic motivation and sports habits on physical self-concept has been added to the already existing about emotional intelligence and physical self-concept

Comment 4

In the material and methods section it is necessary that the authors provide the code of ethics committee.

Response 4.

Authors are grateful for your comment. Code ethics has been provided in material and methods section.

Comment 5.

The results section is sufficiently scientifically sound and precisely detailed. In the discussion section, the authors should point out as a limitation that not all the population studied are older adults.

Response 5.

Author are grateful for your comment. This  limitation has been added.

Comment 6.

As for the references, there are some errors in the adaptation to the regulations. In addition, it could include some more current references on this topic. Please revise.

Response 6.

Authors are grateful for your comment. The manuscript was send in free format. Author have changed format to adapt it to IJERPH.

The following references have been added:

43 Fernandez-Berrocal, P; Extremera, N. A review of Trait Meta-Mood Research. Int. J Psychol Res 2008, 2 (1), 39-67.

50 Martín-Albo, J.; , Núñez, J.L.; Domínguez, E.; León, J.; Tomás, J.M. Relationships between intrinsic motivation, physical self-concept and satisfaction with life: A longitudinal study. J Sports Sci 2012, 30, 4, 337-347,  doi:10.1080/02640414.2011.649776.

51 Padial-Ruz, R.; Pérez-Turpin, J.A.; Cepero-González, M.; Zurita-Ortega, F. Effects of physical self-concept, emotional islation, and family functioning on attitudes towards physical education in adolescents: structural equation analysis. Int. J. Environ. Res. Public Health 2020, 17, 94, doi.org/10.3390/ijerph17010094.

52 Fraguela-Vale, R.; Varela-Garrote, L.; Carretero-García, M.; Peralbo-Rubio, E.M. Basic psychological needs, physical self-concept, and physical activity among adolescents: autonomy in focus. Front. Psychol 2020, 11, 491,  doi: 10.3389/fpsyg.2020.00491.

53 Ramirez-Granizo, I.; Sánchez-Zafra, M.; Zurita-Ortega, F.; Puertas-Molero, P; González-Valero, G; Ubago-Jiménez, J.L. Multidimensional self-concept depending on levels of resilience and the motivational climate directed towards sport in schoolchildren. Int. J. Environ. Res. Public Health 2020,17, 534, doi:10.3390/ijerph17020534.

56 Amado-Alonso, D.; León-del-Barco, B.; Mendo-Lázaro, S.; Sánchez-Miguel, P.A.; Iglesias Gallego, D. Emotional Intelligence and the Practice of Organized Physical-Sport Activity in Children. Sustainability 2019, 11, 1615, doi.org/10.3390/su11061615.

58 World Health Organization. Who guidelines on physical activity and sedentary behaviour. World Health Organization: Geneva, Switzerland, 2020

Round 2

Reviewer 1 Report

The authors addressed most of my suggestions well, but I still have some issues related to emotional intelligence (EI).

The concept of EI was adopted as a trait in this study. But Mayer & Salovey developed a new 4-branch ability model of EI afterwards. They might leave the thought of EI as a trait. In turn, Petrides conceptualized EI as a trait which could fit for the study. It would be better to explain trait EI by Petrides conceptualization rather than Mayer & Salovey.

Author Response

Dear reviewer, 

We appreciate the time taken to review this manuscript and the comments made, which we believe to be critical for producing rigorous and quality research. Below are detailed all the changes made in the original article: ijerph-1104149

Amendments have been made in the original manuscript following reviewer's comments. Changes made are colored to be easily identified.

MODIFICATIONS

REVIEWER 1

Comment 1.

The authors addressed most of my suggestions well, but I still have some issues related to emotional intelligence (EI).

The concept of EI was adopted as a trait in this study. But Mayer & Salovey developed a new 4-branch ability model of EI afterwards. They might leave the thought of EI as a trait. In turn, Petrides conceptualized EI as a trait which could fit for the study. It would be better to explain trait EI by Petrides conceptualization rather than Mayer & Salovey.

Response 1

Authors are grateful for your comments. Mayer & Salovey conceptualization has been replaced for Petrides with this paragraph and references:

“As personality trait, encompasses emotion-related self-perceptions and dispositions concerning how people manage emotions and understand the impact of their emotions on social interactions, and is measured via self-report [43, 44]”.

  1. Petrides, K.V.; Pita, R.; Kokkinaki, F. The location of trait emotional intelligence in personality factor space. Br. J. Psychol 2007, 98, 273–289, doi:10.1348/000712606X120618
  2. Fiorilli, C.; Farina, E.; Buonomo, I.; Costa, S.; Romano, L.; Larcan, R.; Petrides, K.V. Trait Emotional Intelligence and School Burnout: The Mediating Role of Resilience and Academic Anxiety in High School. Int. J. Environ. Res. Public Health 2020, 17, 3058, doi:10.3390/ijerph17093058

Likewise, this paragraph has been deleted from future perspective:

“As well, future works should develop a new self-report measure based on the current Mayer and Salovey´s  model of IE (4-branch), or else adapt the TMMS.”

Reviewer 2 Report

The authors have included the suggestions appropriately. However, some minor changes must be addressed before publication: 

Line 67: Change "cross-section" to "cross-sectional"

Line 100: Please improve grammar of the following phrase "However, in the still few 100 studies on the relationship between physical self-concept and intrinsic motivation, there are so many that they consider it positive [12, 50, 52] as negative [53]" 

Line 106: Remove "Therefore, and" I suggest to start the phrase with "Although.."

Line 111: Change "are:" for "were:". Objectives were stablished in the past.

Line 124: Modify the word "range" as "ranges"

Line 280: The paragraph starts as follows "Our first hypothesis...". Please delete and describe in a full sentence.  

Line 305: Please delete "Our second hypothesis"

Line 374: Change for "mention" instead of "mension"

Author Response

Dear reviewer, 

We appreciate the time taken to review this manuscript and the comments made, which we believe to be critical for producing rigorous and quality research. Below are detailed all the changes made in the original article: ijerph-1104149

Amendments have been made in the original manuscript following reviewer's comments. Changes made are colored to be easily identified.

MODIFICATIONS

REVIEWER 2

Comment 1

The authors have included the suggestions appropriately. However, some minor changes must be addressed before publication:

Line 67: Change "cross-section" to "cross-sectional"

Line 100: Please improve grammar of the following phrase "However, in the still few studies on the relationship between physical self-concept and intrinsic motivation, there are so many that they consider it positive [12, 50, 52] as negative [53]"

Line 106: Remove "Therefore, and" I suggest to start the phrase with "Although.."

Line 111: Change "are:" for "were:". Objectives were stablished in the past.

Line 124: Modify the word "range" as "ranges"

Line 280: The paragraph starts as follows "Our first hypothesis...". Please delete and describe in a full sentence. 

Line 305: Please delete "Our second hypothesis"

Line 374: Change for "mention" instead of "mension"

Response 1

Authors are grateful for your comments. All yours suggestion  has been taken into account and linguistics mistakes solved.  

Likewise, all references to hypothesis has been deleted and paragraph